# Safe infant feeding in healthcare facilities: Assessment of infection prevention and control conditions and behaviors in India, Malawi, and Tanzania

Bethany A. Caruso[1]*, Uriel Paniagua[2], Irving Hoffman[3], Karim Manji[4], Friday Saidi[5], Christopher R. Sudfeld[6], Sunil S. Vernekar[7], Mohamed Bakari[4], Christopher P. Duggan[8,9], George C. Kibogoyo[4], Rodrick Kisenge[4], Sarah Somji[4], Eddah Kafansiyanji[5], Tisungane Mvalo[5,10], Naomie Nyirenda[5], Melda Phiri[5], Roopa Bellad[7], Sangappa Dhaded[7], Chaya K. A.[11], Bhavana Koppad[7], Shilpa Nabapure[12], Saumya Nanda[13], Bipsa Singh[13], S. Yogeshkumar[7], Katelyn Fleming[14], Krysten North[15], Danielle E. Tuller[14], Katherine E. A. Semrau[14,16], Linda Vesel[14], Melissa F. Young[1], for the LIFE Study Group

1 Hubert Department of Global Health, Emory University School of Public Health, Atlanta, Georgia, United States of America, 2 Department of Epidemiology, Emory University School of Public Health, Atlanta, Georgia, United States of America, 3 Institute for Global Health and Infectious Diseases, University of North Carolina at Chapel Hill School of Medicine, Chapel Hill, North Carolina, United States of America, 4 Department of Pediatrics and Child Health, Muhimbili University of Health and Allied Sciences, Dar es Salaam, Tanzania, 5 University of North Carolina Project Malawi, Lilongwe, Malawi, 6 Department of Global Health and Population, Harvard T.H. Chan School of Public Health, Boston, Massachusetts, United States of America, 7 Jawaharlal Nehru Medical College, KLE Academy of Higher Education and Research (Deemed-to-be-University), Belgaum, Karnataka, India, 8 Center for Nutrition, Boston Children's Hospital, Boston, Massachusetts, United States of America, 9 Division of Nutrition, Harvard Medical School, Boston, Massachusetts, United States of America, 10 Department of Pediatrics, University of North Carolina at Chapel Hill School of Medicine, Chapel Hill, North Carolina, United States of America, 11 Bapuji Child Health Institute & Research Centre, Davangere, Karnataka, India, 12 SS Institute of Medical Sciences & Research Centre, Davangere, Karnataka, India, 13 Shri Jagannath Medical College and Hospital, Puri, Odisha, India, 14 Ariadne Labs, Harvard T.H. Chan School of Public Health / Brigham and Women's Hospital, Boston, Massachusetts, United States of America, 15 Brigham and Women's Hospital and Harvard Medical School, Boston, Massachusetts, United States of America, 16 Department of Medicine, Harvard Medical School, Boston, Massachusetts, United States of America

* bcaruso@emory.edu

**Data Availability Statement:** The data presented here are from the Low Birthweight Infant Feeding Exploration (LIFE) study which is filed with

## Abstract

Infants need to receive care in environments that limit their exposure to pathogens. Inadequate water, sanitation, and hygiene (WASH) environments and suboptimal infection prevention and control practices in healthcare settings contribute to the burden of healthcare-associated infections, which are particularly high in low-income settings. Specific research is needed to understand infant feeding preparation in healthcare settings, a task involving multiple behaviors that can introduce pathogens and negatively impact health. To understand feeding preparation practices and potential risks, and to inform strategies for improvement, we assessed facility WASH environments and observed infant feeding preparation practices across 12 facilities in India, Malawi, and Tanzania serving newborn infants. Research was embedded within the Low Birthweight Infant Feeding Exploration (LIFE) observational cohort study, which documented feeding practices and growth patterns to

Clinicaltrials.gov NCT04002908 and Clinical Trial Registry of India CTRI/2019/02/017475. De-identified data (including data dictionaries) are available through the Harvard Dataverse platform under the BetterBirth Dataverse website. This can be found at: https://dataverse.harvard.edu/dataverse/BetterBirthData. The relevant dataset is titled 'Low Birthweight Infant Feeding Exploration (LIFE): In-Facility Observational' The facility profile data is not on Harvard Dataverse due to sensitivity, but all relevant data are accessible in the paper and its appendices.

**Funding:** This work was financially supported by the Bill & Melinda Gates Foundation in the form of a grant (OPP1192260/INV-007326) awarded to KEAS. Under the grant conditions of the Foundation, a Creative Commons Attribution 4.0 Generic License has already been assigned to the Author Accepted Manuscript version that might arise from this submission. This grant also financially supported by the Bill & Melinda Gates Foundation in the form of salaries for BAC, UP, IH, KM, FS, CRS, SSV, MB, CPD, GCK, RK, SS, EK, TM, NN, MP, RB, SD, CKA, BK, S. Nabapure, S. Nanda, BS, SY, KF, DET, KEAS, LV, and MFY. The specific roles of these authors are articulated in the 'author contributions' section. No additional external funding was received for this study. The Bill & Melinda Gates Foundation reviewed the study design, but had no role in data collection, management, analysis, interpretation, writing of the manuscript, or the decision to submit manuscripts for publication. The findings and conclusions contained within are those of the authors and do not necessarily reflect positions or policies of the Bill & Melinda Gates Foundation.

**Competing interests:** The authors have read the journal's policy and have the following competing interests: BAC, KM, KEAS, KF, LV, DET, and CRS report funding from the Bill & Melinda Gates Foundation outside the submitted work. BAC reports funding from the National Institutes of Health outside the submitted work. CPD reports editorial duties with American Society for Nutrition and royalties from People's Medical Publishing House (PMPH USA, Ltd.) outside the submitted work. CPD reports royalties from Wolters Kluwer Health (UpToDate, Inc.) outside the submitted work. This does not alter our adherence to PLOS policies on sharing data and materials. There are no patents, products in development or marketed products associated with this research to declare.

inform feeding interventions. We assessed WASH-related environments and feeding policies of all 12 facilities involved in the LIFE study. Additionally, we used a guidance-informed tool to carry out 27 feeding preparation observations across 9 facilities, enabling assessment of 270 total behaviors. All facilities had 'improved' water and sanitation services. Only 50% had written procedures for preparing expressed breastmilk; 50% had written procedures for cleaning, drying, and storage of infant feeding implements; and 33% had written procedures for preparing infant formula. Among 270 behaviors assessed across the 27 feeding preparation observations, 46 (17.0%) practices were carried out sub-optimally, including preparers not handwashing prior to preparation, and cleaning, drying, and storing of feeding implements in ways that do not effectively prevent contamination. While further research is needed to improve assessment tools and to identify specific microbial risks of the suboptimal behaviors identified, the evidence generated is sufficient to justify investment in developing guidance and programing to strengthen infant feeding preparation practices to ensure optimal newborn health.

## Introduction

Infants need to receive care in environments that limit their risks and nurture their growth, particularly those at increased risk. With over 80% of all births worldwide occurring in a healthcare facility (HCF) [1], the quality of healthcare environments is critical. Evidence demonstrates that the existence and level of neonatal hospital care available can impact the mortality of LBW infants [2,3]. Yet, in sub-Saharan Africa and South Asia, where approximately 75% of LBW infants are born [4], facilities have been shown to lack readiness (e.g., resources) to care for small and sick newborns, who are at greatest risk [5]. Further, while there has been a push for institutional childbirth to prevent avoidable maternal and infant mortality during delivery [6], the facilities themselves may pose a risk to infants once they are born due to unhygienic conditions [7]. The burden of healthcare-associated infections (HCAIs) is particularly high in low-income settings [8], where hospital-born infants are at increased risk compared to hospital-born infants in high income settings [9].

Infection from pathogen exposure may be particularly harmful to low birthweight (LBW) infants who have less well developed immune systems, making them more susceptible to infection and impacts on future growth and development [10]. Globally, an estimated 15% of infants were born low birthweight (<2.5kg) in 2015, a condition responsible for an estimated 60–80% of neonatal deaths [1]. Compared with infants of a birthweight >2.5 kg, LBW infants are at greater risk of morbidity, nosocomial infection, developmental delays, growth deficits [1,11–15], and feeding challenges [16,17]. The vulnerabilities faced by LBW infants underscore their need to receive care in environments that not only support their growth and development, but also limit pathogen exposure.

Inadequate water, sanitation, and hygiene (WASH) at healthcare facilities may contribute to the risk of pathogen exposure and resultant HCAIs, increase the spread of antimicrobial resistant bacteria, undermine the quality of care being provided and the cleanliness and infection prevention and control (IPC) measures present, and compromise the dignity and satisfaction experienced by patients and healthcare workers [7,18–21]. Yet despite its importance, millions of healthcare facilities worldwide lack basic WASH services [18,22]. The 2019 global baseline report by the WHO/UNICEF Joint Monitoring Programme for Water Supply, Sanitation and Hygiene (JMP) highlighted that an estimated 10% of healthcare facilities had no

sanitation services, 16% had no hygiene services, specifically handwashing facilities at points of care and water and soap at toilets, and 26% lacked on premises access to water from an improved source, which by design or construction 'have the potential to deliver safe water' [18].

Access to WASH services does not guarantee HCFs are inherently safe [22]; key behaviors are also required [19,23–25]. Handwashing is particularly important and effective at reducing pathogen exposure, specifically before patient contact and aseptic tasks, and after contact with patients, patient surroundings, and body fluids [26,27]. Additionally, as has been noted elsewhere, thousands of caregiving behaviors together create an enabling microenvironment for the optimal growth and development of each child, including approximately 3000 feeds by 24 months [28]. Feeding preparation itself is a complex multi-step process [29], involving hand washing as well as proper cleaning, drying, and storage of feeding implements. Each of these behaviors and interactions has the potential to introduce pathogens, cause infection, and negatively impact health, particularly among sick and vulnerable infants who are at increased risk. Therefore, there is a need to not only assess and improve the WASH conditions in the HCFs in which newborns receive care, but to also assess and improve the specific behaviors related to feeding. While tools exist to assess and guide improvements in WASH in HCFs [30–33], these primarily focus on assessing available HCF resources and infrastructure. Context-specific tools for assessing infant feeding behaviors related to WASH and infection prevention and control are needed.

The primary aims of this study were to 1) assess the WASH environments of healthcare facilities serving newborn infants in low- and middle- income countries (LMICs); and 2) identify potential opportunities for pathogen introduction during infant feeding by observing feeding preparation behaviors across facilities.

## Methods

### Study design

This study was a component of the Low Birthweight Infant Feeding Exploration (LIFE) study, an observational cohort study that aimed to document the current feeding practices and growth patterns among LBW infants in LMICs in order to inform feeding interventions. Additional information about the cohort study methods can be found in the protocol [34], in the paper documenting infants' feeding practices, growth patterns, and risk factors for growth outcomes at six months [35], and in a paper describing facility-based care for moderately low birthweight infants in these settings [36]. This paper reports findings related to WASH and IPC from two data streams: the facility needs assessment and the feeding preparation observation data as part of the in-facility observational cohort.

### Study setting

The LIFE study was conducted in 12 secondary and tertiary level facilities (i.e., facilities where patients are referred by their primary care provider for more specialized service) in four urban sites across three countries: (1) Karnataka and (2) Odisha states in India; (3) Lilongwe, Malawi; and (4) Dar es Salaam, Tanzania. Facilities were selected based on their delivery volume, capacity to care for LBW infants in the first days of life, and the willingness of facility leadership to engage in the study; both public and private facilities were included [34].

### Data collection and analysis

**Facility needs assessment.** Trained data collectors used standardized tools across all facilities to record the structural, human resource, equipment, and service inputs present for

mothers and their newborns at each facility. Data was collected via direct observation and in consultation with staff for confirmation. This paper specifically reports the WASH-related resources and policies documented in the facility assessments; a forthcoming paper documents the availability of other resources in the facilities (e.g., medications, milk expression tools, anthropometric measuring equipment) that support care for LBW infants. Questions used to assess facility-level water and sanitation are among the 'core questions' used for global monitoring of water and sanitation in healthcare facilities by the WHO/UNICEF Joint Monitoring Programme for Water Supply, Sanitation and Hygiene for global monitoring [37].

We calculated the proportions of healthcare facilities with policies related to infant feeding and access to WASH services. For WASH, we specifically classified whether the water and sanitation sources a facility have are considered 'improved'. Improved water sources are those that 'by nature of their design and construction, have the potential to deliver safe water' and include piped water, boreholes or tubewells, protected dug wells, protected springs, rainwater, and packaged or delivered water. Improved sanitation facilities 'are those designed to hygienically separate excreta from human contact' and include flush/pour flush to piped sewer system, septic tanks or pit latrines; ventilated improved pit latrines, composting toilets or pit latrines with slabs [37].

**In-facility feeding preparation observation.** Data collectors conducted observations to assess the quality of WASH conditions and practices related to feed preparations. There is no standardized tool for assessing infant feed preparation in healthcare facilities. As such, we created feeding preparation observation prompts based on relevant guidance from the Joint Working Group of the Healthcare Infection Society & Infection Prevention Society on the decontamination of breast pump milk collection kits and related items at home and in hospital [38]. Given that the guidance was created for and informed by practitioners in the United Kingdom, observation prompts were added or adapted to fit the context and research scope. For example, Guideline 1.2 emphasizes the importance of 'good microbiological quality water' where mothers express milk and for rinsing of feeding-related implements. Microbial water quality assessment was not possible, thus questions about water availability in the preparation room and treatment serve as proxies. Treatment was assessed because having an improved water source on premises does not guarantee water is safe, only that its construction and design has the ability to deliver safe water. The measures and the specific guidance informing the measure (as relevant) are summarized in Table 1 and visually depicted in Fig 1. Though there is no specific guideline for glove use, we added a prompt to record glove use. Importantly, glove use is not a substitute for handwashing in healthcare settings [39,40], though it may be perceived as a substitute. Thus, our intention was to understand if glove use was practiced and, if so, handwashing was also carried out.

To fill out the tool, data collectors observed a facility staff member preparing a feed for an infant who was not receiving milk directly from the breast. The feed could be for any baby in the facility, not only those enrolled in the study cohort. Follow-up questions were asked in instances of uncertainty (e.g., if and how water is treated). Observations were carried out by members of the facility-based data collection teams, which were comprised of approximately 6–12 trained nurses in each facility, with variability by site and over time. The facility-based data collection teams were at the facilities for the duration of the enrollment period (13 September 2019 and 27 January 2021) to engage in cohort-related research activities, including enrolling LBW infants in the cohort or carrying out observations of care received by enrolled infants at the facility. For the feed preparation observations, data collectors were advised to observe and collect data on five feeding preparations per site, when possible, ideally on different days for variability. They were instructed to carry out observations only if not engaged in other cohort research activities for which they were also responsible (e.g., cohort enrollment,

**Table 1. Feed preparation observation survey questions and supporting evidence from the joint working group of the healthcare infection society & infection prevention society [38].**

| Question | Response Choices | Informing Guideline |
|---|---|---|
| *Water Source* | | |
| a. Does the preparation room have a water source? | • Functional piped water into room*<br>• Water stored in room from outside source<br>• No water source in room<br>• Don't know | 1.2 |
| b. How is the water source treated? | • N/A - Bottled/sachet water*<br>• Boiled prior to use*<br>• Filtered prior to use*<br>• Treated with chlorine/bleach prior to use*<br>• Other treatment<br>• Not treated<br>• Don't know | 1.2 |
| *Hand Hygiene* | | |
| c. Does the person preparing the feed wash their hands before starting any feed preparation activities? | • Yes, with soap and water*<br>• Yes, with only water<br>• No<br>• Not observed | 2.2.1 |
| *Feeding Implement Hygiene* | | |
| d. Were any of the feeding supplies cleaned immediately before use? | • Yes, feeding supplies washed with WATER AND SOAP by hand immediately before use*<br>• Yes, feeding supplies washed with WATER ONLY by hand immediately before use<br>• Yes, feeding supplies washed with WATER AND SOAP by dishwasher machine immediately before use<br>• No, feeding supplies not washed immediately before use<br>• Not observed | 2.1.1 |
| e. What is used to wash the feeding supplies prior to use? | • New/not previously used sponge/brush*<br>• Previously used sponge/brush, sterilized*<br>• Previously used sponge/brush, NOT sterilized<br>• Other<br>• Not observed | 2.1.1 |
| f. Were the feeding supplies dry before use? | • Yes, feeding supplies dried with cloth before use<br>• Yes, feeding supplies dried with paper/disposable towel before use*<br>• Yes, feeding supplies taken from rack where air dried before use*<br>• Yes, feeding supplies dry but method not observed<br>• No, feeding supplies not fully dry before use (some or all still wet)<br>• Not observed | 2.3.6 |
| g. Are the feeding supplies cleaned after use? | • Yes, feeding supplies washed by hand with WATER and SOAP after use*<br>• Yes, feeding supplies washed by hand with WATER ONLY after use<br>• Yes, feeding supplies washed in dishwasher machine after use*<br>• No, feeding supplies NOT immediately washed after use<br>• No, feeding supplies discarded after use<br>• Not observed | 2.3.1<br>2.3.4<br>2.4 |

(*Continued*)

**Table 1.** (Continued)

| Question | Response Choices | Informing Guideline |
|---|---|---|
| h. Are feeding supplies dried before storage or next use? | • Yes, feeding supplies dried as part of drying cycle in dishwasher machine prior to storage or next use*<br>• Yes, feeding supplies dried with cloth before storage or next use<br>• Yes, feeding supplies dried with paper/disposable towel before storage or next use*<br>• Yes, feeding supplies air drying in rack for before storage or next use*<br>• No, feeding supplies NOT dried prior to storage or next use.<br>• Not observed | 2.3.6 |
| i. Are feeding supplies placed in sterile bags/products after cleaning to sustain cleanliness for next use? | • Yes*<br>• No<br>• Not observed | 3.3.2 |
| j. Where were feeding supplies stored when not in use? | • Out in open<br>• In cabinets*<br>• On a tray covered with clean cloth*<br>• Sterile container (bucket with lid)*<br>• Sterile Steel Box*<br>• Not observed | 2.3.8 |

*optimal response according to guidance.

other infant-care observations). They did not record who the prepared feed was intended for and, as a result, data are not linked to a specific infant. The aim was to simply gain insights about facility practices and procedures when preparing an infant feed.

We calculated the frequency of the various WASH-related practices for feed preparations overall and by facility. For each observation, we also determined which practices carried out as part of the feeding preparation were optimal and which were suboptimal based on the

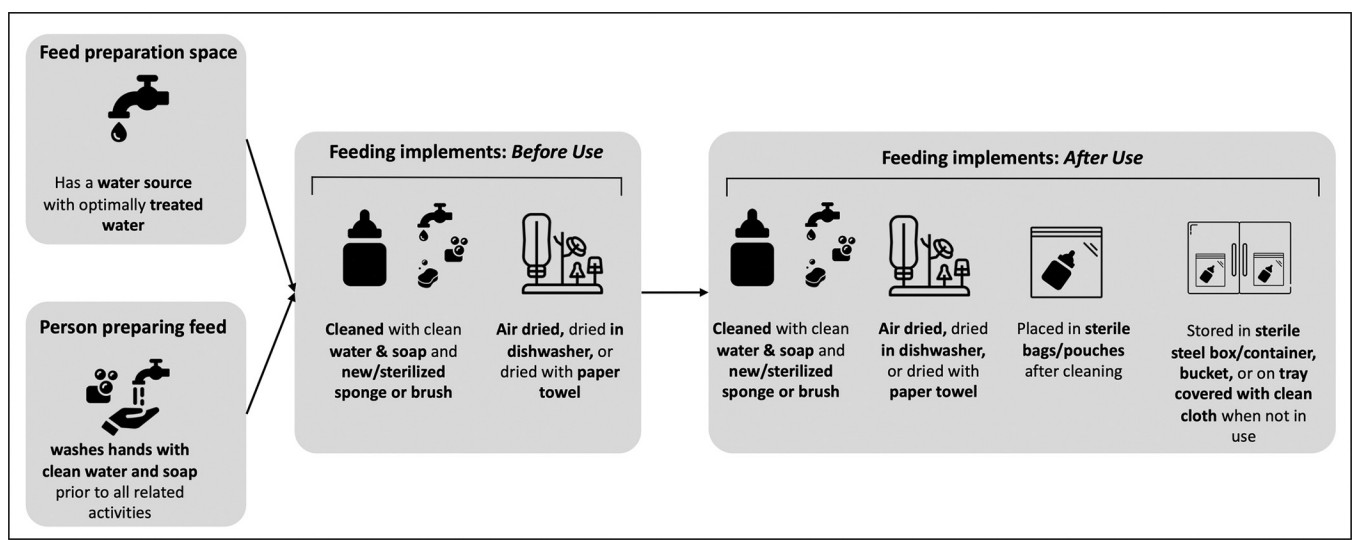

**Fig 1. Conceptual framework of optimal feeding preparation conditions and practices in healthcare facilities based on the joint working group of the healthcare infection society & infection prevention society [38].**

guidelines. For observations in which both an optimal and suboptimal response were observed, the suboptimal category was assigned (e.g., if a cloth towel (suboptimal) and a paper towel (optimal) were used to dry a feeding implement, it would be considered suboptimal).

We employed a "traffic-light" classification scheme, as is in use by the Child Health and Wellbeing dashboard [41], to visually depict whether or not facility conditions and each feeding preparation practice observed was optimal or suboptimal. This approach facilitates assessment of the number of risks within one preparation observation, as well as trends within and between facilities, and at a particular study site. Tables and figures were de-identified; facilities were randomly labeled A-L to maintain confidentiality, though are sorted by study site. All analyses were carried out using R version 4.1.3.

## Ethics

Eleven ethics committees in India, Malawi, Tanzania and the USA approved this study: (1) India Health Ministry's Screening Committee with Indian Council of Medical Research acting as its secretariat (2019–2674); (2) Directorate of Health and Family Welfare Services, Government of Karnataka, which also covers investigators at Women and Children Hospital, Davangere and Chigateri General District Hospital, Davangere (NHM/SPM/04/2019–20); (3) Institutional Ethics Committee of KLE Academy of Higher Education and Research which also covers investigators at JN Medical College, Belagavi and KLES Dr Prabhakar Kore Hospital & Medical Research Center, Belagavi (KAHER/IEC/2019–20/D-2760); (4) Institutional Ethics Review Board of SS Institute of Medical Sciences and Research Centre (IERB/200/ 2019); (5) Institutional Ethics Committee of JJM Medical College (JJMMC/IEC-01/2019), which also covers investigators at Bapuji Child Health Institute and Research Centre, Davangere, Women and Children Hospital, Davangere and Chigateri General District Hospital, Davangere; (6) Research and Ethics Committee, Directorate of Health Services, Odisha state, which also covers investigators at City Hospital Oriya Bazar, Cuttack (155/PMU/187/17); (7) Institutional Ethical Committee, Sriram Chandra Bhanja Medical College, Cuttack (7188); (8) Malawi National Health Sciences Research Committee (NHSRC2019/Protocol19/03/ 2250-UNCPM 21905); (9) Tanzania National Institute of Medical Research (NIMR/HQ/R.8a/ Vol.IX/3126); (10) Muhimbili University of Health and Allied Sciences (DA.282/298/01.C/); and (11) Harvard T.H Chan School of Public Health (IRB10-0282) which also covers investigators at Boston Children's Hospital, Brigham and Women's Hospital, Emory University, PATH and University of North Carolina. Additional details about the ethical approvals are provided in the published protocol [34]. Verbal consent was obtained from facility leadership.

## Results

### Facility-level water and sanitation services and infant feeding-related policies and procedures

The facility assessment was completed in all 12 participating facilities (Table 2). Overall, the facilities had robust water and sanitation services. All 12 had a drinking water source either piped inside the building (9/12; 75%) or received packaged/bottled water/water from a dispenser (3/12; 25%), sources that are considered 'improved' by the JMP. Among the facilities with a drinking water source piped inside the building, 7/9 (78%) reported treating the drinking water through filtration, boiling, ultraviolet disinfection, or reverse osmosis and one reported no treatment of drinking water (one had missing data). All the reported methods of water treatment are considered optimal. The facility that reported boiling their drinking water indicated that the water was only sometimes boiled when used for infant formula; otherwise, it

**Table 2. Facility-level water and sanitation services, and feeding-related procedures and practices.**

| | Overall (N = 12) |
|---|---|
| **Facility toilet/latrine type** | |
| Flush/Pour-flush toilet to sewer connection | 11 (92%) |
| Flush/Pour-flush toilet to tank or pit | 1 (8%) |
| **Drinking water source** | |
| Piped Supply Inside the Building | 9 (75%) |
| Packaged/Bottled Water/Dispenser | 3 (25%) |
| **Drinking water availability at time of survey** | |
| Yes | 11 (92%) |
| Unknown | 1 (8%) |
| **Written policies & procedures[1]** | |
| Infant formula preparation | 4 (33%) |
| Preparation of expressed breastmilk for infant feeding | 6 (50%) |
| Cleaning, drying, and storage of implements used for milk expression, feed prep, and infant feeding | 6 (50%) |
| **Drinking water treatment[2]** | |
| Yes | 7 (78%) |
| No | 1 (11%) |
| N/A or unknown | 1 (11%) |
| **Drinking water treatment method[3]** | |
| Filtration | 2 (29%) |
| Boiling | 1 (14%) |
| Ultraviolet disinfection | 2 (29%) |
| Reverse osmosis | 2 (29%) |
| **Person(s) involved in preparation of infant feeds[1]** | |
| Physicians - Pediatricians | 1 (8%) |
| Physicians - House staff or Medical officers | 1 (8%) |
| Nurses | 9 (75%) |
| Mothers | 9 (75%) |
| Other family members | 3 (25%) |
| Facility kitchen help | 1 (8%) |
| **Location where preparation of infant feed takes place[1]** | |
| Where baby is staying (Nursery, NICU) | 11 (92%) |
| Room where mother is staying | 3 (25%) |
| Designated food preparation room/area | 1 (8%) |

1. Multiple responses possible.

2. Among the 9 facilities that reported a drinking water source piped inside the building.

3. Among the 7 facilities that reported drinking water.

was used straight from the tap for other purposes. Drinking water was available in 11/12 (92%) facilities at the time of the survey (availability was unknown in the remaining facility). All the facilities had a flush/pour-flush toilet, with 11/12 having a sewer connection and the other having a tank or pit. The sanitation conditions are considered 'improved' by the JMP [37].

Policies and procedures related to infant feeding practices were not available in all facilities. Only 6/12 (50%) reported having written policies/procedures regarding preparation of expressed breastmilk for infant feeding, 6/12 (50%) reported having written policies/procedures for cleaning, drying, and storage of implements used for milk expression, feed

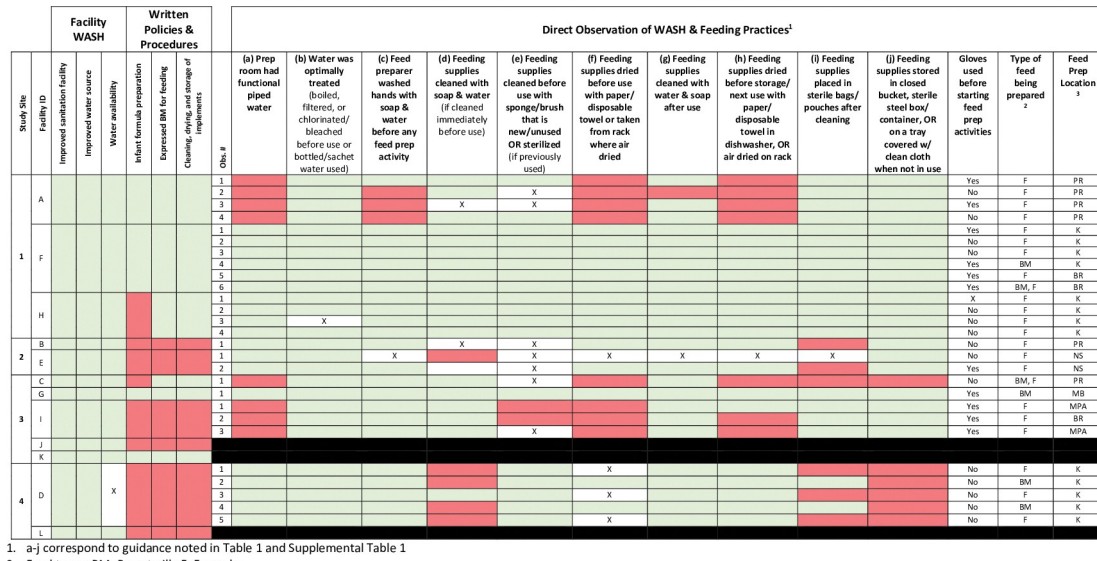

**Fig 2. Traffic light visual identifying optimal and suboptimal practices within and across facilities based on direct observations.**

preparation, and infant feeding, and 4/12 (33%) reported having written policies/procedures for infant formula preparation. Four (25%) of the facilities reported policies and procedures for all three types of practices queried (expressed breastmilk; cleaning drying, and storage of implements; and formula preparation), while 6/12 (50%) reported having no policies and procedures for any of these practices (Fig 2).

At the facility level, multiple types of individuals were reported to be involved in the preparation of infant feeds, including nurses (9/12; 75%), mothers (9/12; 75%), other family members (3/12; 25%), pediatricians (1/12; 8%), house staff/medical officers (1/12; 8%), and kitchen help (1; 8%). Almost all facilities reported infant feeds to be prepared where the infant was staying (11/12; 92%), though other locations were also used.

### Direct observations of feed preparation practices

A total of 27 feed preparation observations were completed across 9 of the 12 (75%) facilities participating in the LIFE study for 10 practices with optimal guidelines (270 total) (Fig 2). Among those, only one feeding observation was completed for three of the facilities (33%) while multiple observations (range: 2–6) were conducted across the remaining six (66%). Whether or not the practice observed was considered optimal according to the guidelines is depicted in Fig 2 (specific responses/observations related to feeding assessment questions are reported in S1 Table by facility). Of the 27 observations, only 10 (37%) had each practice observed carried out optimally, meaning that 17 (63%) of the observations had at least one suboptimal behavior. Among 270 practices assessed, 46 (17.0%) practices were carried out suboptimally (18 (6.6%) had missing data).

Water treatment and the cleaning of feeding supplies after use were two practices done well across facilities and feeding assessments. Specifically, an optimal water treatment practice was reported across all facilities (9/9) and across almost all observations (26/27; 96%) (1 observation with missing data), despite a suboptimal preparation room water source (water that is

stored in the room from an outside source) being reported across three facilities. Similarly, across almost all observations (25/27; 93%), feeding supplies were reported to be cleaned after each use.

Suboptimal behaviors were recorded related to cleaning, drying, and storing of feeding implements. Two observations from the same facility recorded a sub-optimal method for washing feeding implements prior to use (e.g., not using a new or sterilized sponge or brush). Suboptimal methods were recorded for 8/27 (30%) observations related to drying before feeding implement use; for 7/27 (26%) related to drying after implement use, and in 3/27 (11%) for practicing neither storage behavior optimally.

Hand washing prior to feeding preparation was suboptimal in 11% (3/27) of observations, all occurring in a single facility. While gloves were reported to be in use for almost half of the assessments (11/27; 41%), handwashing is still required when using gloves and did not occur in one instance when gloves use was observed. In two observations from the same facility (A), hands were neither washed nor gloved.

## Discussion

We assessed the WASH services of 12 healthcare facilities across three countries that care for vulnerable newborns, and observed feeding preparation practices across nine of those facilities to identify potential opportunities for pathogen introduction during newborn feeding. Using standardized measures for monitoring healthcare facility-level water and sanitation, we found that all facilities had access to improved water sources and sanitation facilities. However, we also found that nearly one fifth of the practices observed within facilities were suboptimal given their potential to expose newborns to pathogens. While there is a need for more extensive research to identify the specific risk of pathogen introduction and infection associated with these suboptimal behaviors, this research demonstrates that improvements to feeding preparation practices are warranted.

All 12 facilities assessed in this study had improved water and sanitation facilities, which are necessary for infection prevention and control, but not sufficient, particularly if services are not functional, accessible, designed appropriately, or located where needed [22,42]. As Burki (2019) notes, access to basic WASH, though critical, does 'not necessarily equate to safe services' [22]. Our research supports Burki's point, having identified various types of unsafe practices related to newborn feeding within the facilities despite facility-level WASH access. In other words, assessing water and sanitation access alone as proxies for assessing facility-level resources to prevent and control infection is insufficient and potentially misleading. For example, this work further demonstrates that it is critical for a water source to not only be available at the facilities, but where hygiene behaviors are to occur. We found that handwashing did not occur before feeding preparation in 3/27 (11%) observations, all clustered in just two healthcare facilities. In three instances, we observed that water was not available in the room where the feed was being prepared, potentially serving as the primary barrier for handwashing in those instances. Having the necessary resources—including water, soap, and disposable towels for drying—within reach in food preparation areas creates the ideal environment for handwashing prior to handling food and implements [29]. Of course, not all facilities will have the resources to make water, soap, and disposable towels in each room, but recommendations should still be made clear, with options for less resourced areas (e.g., air drying is also a safe drying strategy; cloth towels should be avoided). For the other observations where handwashing did not occur, water was observed to be available in the room, clearly showing that resource availability alone is not enough to ensure optimal behaviors. Additionally, given that we observed the use of gloves, without handwashing prior, the availability of resources may

hamper proper hygiene. Gloves are not a replacement for hand hygiene as they can break and expose contaminated hands [40,43].

Both resource and context-specific behavioral guidance and behavior change strategies are needed to ensure optimal hygiene behaviors related to child feeding to prevent pathogen introduction and infection, as has also been observed in household contexts [44]. For example, general guidance should be clear about the need to wash hands before handling feeding implements, but may be unfeasible if overly prescriptive by requiring a water source with a tap in each health facility room. Context-specific behavior change programming to improve handwashing should be informed by resources available as well as the identified barriers and motivators to performing hand hygiene in a specific setting. Programs designed to improve handwashing behaviors in healthcare settings serving infants have proven effective. Having observed increased incidence of invasive *Candida* infections (ICIs) among preterm infants in the NICU of a Chinese hospital, researchers undertook a retrospective study to evaluate different prevention measures and found ICI to be significantly less frequent in an intervention that integrated hand hygiene education, management, and supervision with prophylactic intravenous fluconazole (used to prevent and treat fungal infections) compared to the control group or providing prophylactic intravenous fluconazole alone [27]. Additionally, a quality improvement tool, like the World Health Organization (WHO) Safe Childbirth Checklist [45], could be designed to guide and ensure adherence to optimal practices for infection prevention and control related to infant feeding, with coaching or other strategies employed to facilitate uptake and improvement of practices [46]. Investment is needed, however, to ensure that programs and policies developed reach those they are intended to reach. Recent research in Uganda found that only 44% of the 59 HCFs observed had WASH and IPC guidelines and only 42% had trained staff on WASH-related issues, and the authors concluded that leadership, financing, monitoring and evaluation, proper training, and accountability were crucial [47]. As we also found that having policies in place did not guarantee optimal behaviors, it is clear that strategies needed go beyond policy provision alone to ensure that all involved in feeding in HCFs are sufficiently trained and that optimal conditions and behaviors are sustained. For example, while nurses and nurses assistants were involved in the feed preparations that we observed, multiple types of individuals were reported to prepare feeds at the facility level, including physicians, kitchen help, mothers, and other family members. Optimal feeding preparation in facilities by all involved is needed not only for child health, but to serve as an opportunity to teach all parents and all infant caregivers how to prepare feeds optimally for when infants are no longer in HCFs.

Novel to this study is the observation of behaviors beyond hand hygiene. We created and used a guidance-informed tool for within-facility WASH and IPC risk assessment related to infant feed preparation to assess a more diverse subset of behaviors beyond hand hygiene. We found that proper cleaning of feeding implements before and after use was not universal, and that optimal drying and storage behaviors were a particular challenge. Proper drying, whether air drying, or using paper towels or an industrial washer, ensures residual water does not remain on implements and allow for bacterial and fungal growth; improper drying with a reusable cloth may remove water but re-contaminate feeding implements [38]. Proper storage, in sterile pouches or containers, in cabinets, or even covered with a cloth, is necessary to maintain cleanliness and prevent recontamination [29]. Proper drying and storage were particularly problematic in facilities that reported having no facility-level policies or procedures related to cleaning, drying, and storage, but not exclusively. Further research should involve microbial assessments and other methods of evaluating cleanliness of healthcare environments [48,49], for example swabs of hands, implements, and surfaces—which were not assessed in this study but could pose a risk [19]. Assessment of water quality should also be carried out to determine

if water being used to clean implements or mix formula is contaminated [38]. Microbial assessments can be carried out alongside observations to determine the extent to which the varying feeding preparation-related behaviors prevent or enable the introduction of pathogens to newborns.

This research adds to a growing body of work that has used observational research in health facilities to identify potential opportunities for pathogen introduction from birth through the postnatal period, though to our knowledge this is the first study to focus specifically on infant feeding. Research in Nigeria [24] and Tanzania [23] both observed poor hand hygiene compliance among healthcare workers prior to aseptic birth-related procedures, as well as risks of hand recontamination among those who did wash hands. Another study in Nigeria found that adequate hand hygiene was practiced among only 1% of potential hand hygiene opportunities observed at the healthcare facility and within 6 hours of the newborn returning home [25]. Together, these observational studies and the present research collectively demonstrate the need for improved hand hygiene along the newborn continuum of care, particularly in healthcare settings. Indeed, intensive hand hygiene interventions with healthcare workers have been successful in preventing infections among vulnerable infants in China [27], and similar interventions need to be adapted to be appropriate for the diverse contexts and resources available in LMIC settings.

## Strengths and limitations

We created a novel, guidance-informed tool to assess infection risk related to infant feeding that allowed us to present data by observation, allowing identification of trends by and across facilities and study sites. The guidance informing the tool, though informative, was created for high-income settings and was specific to decontamination of breast pump parts in healthcare settings. Given that feeding and feed preparation is a complex, multi-step process [29], further work should be done to improve both the tool and conceptual model to ensure that they both comprehensively capture all conditions and behaviors relevant to infant feeding in healthcare settings. We already recognize changes needed in the tool we created. Specifically, while we ask if hand washing and feeding implement cleaning is carried out with soap, we did record if soap or cleaning agents were even available where the behaviors were taking place. Availability of soap may be expected in high resource settings, though is not guaranteed, particularly in low resource settings. Future work should record the availability of hand and implement cleaning agents given that their absence prevents optimal behaviors and requires critical facility-level action. Unstructured observations across multiple settings could be carried out as a first step to inform further modifications to the tool [50]. Adaptation of the tool for use with applications like LiveTrak (Stanford University, CA; open-source link: https://github.com/chrisdembia/LiveTrak) could also be done to track the feeding preparation event in real time, including the order of behaviors and if and when possible (re)contamination events occur [50]. We report whether a facility representative indicated the existence of policies or procedures related to infant feeding, but were not able to carry out a content assessment of those policies to assess quality.

Findings may not be generalizable to other locations as the facilities engaged are in urban settings, and there was also the potential for observer bias. Further, by nature of their eligibility for inclusion in the broader cohort study, the facilities themselves had the capacity to care for LBW infants in the first days of life and were willing to engage in the study, thus may have been fairly well-equipped compared to other facilities in the area. Data was collected at one point in time, potentially masking the impact of seasonal changes or power access that may influence water access. Forthcoming work from this research team will

provide insights about other facility-level resources that could enable facilities to be resilient against or at risk during such challenges, including access to solar power or generators [34]. As with any data collection involving observation, Hawthorne effect (those being observed changing behavior due to awareness of being observed) is always a potential limitation. Those preparing feeds could have changed behavior, though we assume they would be more likely to perform a behavior optimally when observed. As such, if the Hawthorne effect is a factor in this study, we most likely under-observed sub-optimal behaviors, resulting in conservative estimates of the proportion that were suboptimal. We were unable to complete our target number of observations across all twelve facilities as planned because data collection teams were engaged in other research activities in those locations that were prioritized. However, though 27 observations were completed across nine facilities in three countries, providing data on 270 total practices. The tool has proven useful in identifying opportunities for practice improvement, within and across facilities and could be modified for use in other contexts.

Finally, improvements to feeding behaviors are critical in healthcare settings, though no less important when infants are at home. This research focused on behaviors in healthcare settings because infants are increasingly delivered in a facility where they may be at risk of HCAIs and may be in particular need of feeding assistance if LBW. Those working in facilities could be models for mothers and other caregivers. As such, they have an opportunity to model and teach optimal behaviors so they may be taken up beyond the healthcare setting. Further work, however, is needed to encourage optimal feeding strategies for infants in the home that are both nutritious and safe, and interventions like those in Kenya focusing on 'mealtime' safety may be a useful guide [51,52].

## Conclusion

This research has contributed insights about potential IPC needs, including the role of enhanced WASH-related resources and behaviors in healthcare settings and priorities for improving practice (Table 3). While there remains limited information about the role of WASH in healthcare acquired infections [21], this work identifies specific risks related to newborn feeding behaviors, despite the availability of WASH infrastructure. Further work is needed to improve tools for assessment and to identify specific microbial risks related to behaviors. However, the evidence we generated is sufficient to justify improvements in WASH resourcing and practice to ensure newborn health is not compromised.

**Table 3. Priority areas for improving safety of infant feeding in healthcare facilities.**

• **Standardized protocols for healthcare settings** to improve and ensure safety of infant feeding preparation activities across the *full spectrum of opportunities for infection* (water source/treatment → handwashing → feeding supplies properly washed/dried/stored → infant milk/formula properly provided/stored).
• **Model and teach safe infant feeding techniques to all engaged in infant feed preparation,** including mothers and family members, to ensure proper behaviors are practiced in facilities, but are also learned by those who may be involved after leaving the facility.
• **Investment and evaluation of context-specific behavioral guidance and behavior change strategies** to ensure optimal hygiene behaviors to prevent pathogen introduction and infection are adopted, particularly those related to child feeding.
• **Enhanced research to further understand**
  ○ **additional behaviors and conditions that may contribute to pathogen risk;**
  ○ **the extent of pathogen risks related to identified infant feeding behaviors** in healthcare settings, including the extent of contamination of implements used for feeding and environments where clearing takes place, and of water used to clean infant feeding implements or mix formula.

## Supporting information

**S1 Table. Direct observations of WASH practices across and within facilities.**
(DOCX)

**S1 Text. PLOS' questionnaire on inclusivity in global research.**
(DOCX)

## Acknowledgments

The authors would like to thank clinical leadership and staff at all study facilities for their partnership, support, and contribution to this work, all data collectors and study staff for conducting study activities, with noted thanks to Pavan Dhamone and Nikhil Naik from the data team and the entire study nursing teams at the India sites.

Life Study Group includes:

*Davanagere*, *Karnataka*, *India*: Dr Shivaprasad Goudar, Dr.Veena Herekar, Dr Shridevi Metgud, Dr Umesh Charintamath, Dr Manjunath Somannavar, Dr Chandrashekhar C Karadiguddi, Geetanjali Mungarwadi, Dr Guruprasad Goudar, Dr. Koujalagi M B., Dr Saroja Kamatar, Dr Latha, Dr Mruthyunjay, Dr Suresh Kumar, Dr. Latha G. Shamanur, Dr. Varun B Kusagur, Dr. Siddhartha E S, Dr. Venna G S, Dr. Sudha C Patil.

*Cuttack*, *Odisha*, *India*: Dr Jnanindranath N Behera, Dr.Rashmita B Nayak, Dr. Leena Das, Dr. Sujata Misra, Dr. Sanghamitra Panda.

*Lilongwe*, *Malawi*: Albans Msika, Kingsly Msimuko.

*Dar es Salaam*, *Tanzania*: Dr. Robert Moshiro, Dr. Nahya Salim, Dr. Abraham Samma, Sr. Veneranda Ndensangia, Dr. Kristina Lugangira, Sr. Juliana Mghambha.

## Author Contributions

**Conceptualization:** Bethany A. Caruso, Friday Saidi, Krysten North, Danielle E. Tuller, Katherine E. A. Semrau, Linda Vesel, Melissa F. Young.

**Data curation:** Linda Vesel.

**Formal analysis:** Bethany A. Caruso, Uriel Paniagua, Melissa F. Young.

**Funding acquisition:** Katelyn Fleming, Danielle E. Tuller, Katherine E. A. Semrau, Linda Vesel.

**Investigation:** Irving Hoffman, Karim Manji, Friday Saidi, Christopher R. Sudfeld, Sunil S. Vernekar, Mohamed Bakari, George C. Kibogoyo, Rodrick Kisenge, Sarah Somji, Eddah Kafansiyanji, Tisungane Mvalo, Naomie Nyirenda, Melda Phiri, Roopa Bellad, Sangappa Dhaded, Chaya K. A., Bhavana Koppad, Shilpa Nabapure, Saumya Nanda, Bipsa Singh, S. Yogeshkumar.

**Methodology:** Bethany A. Caruso, Katherine E. A. Semrau, Linda Vesel, Melissa F. Young.

**Project administration:** Katelyn Fleming, Danielle E. Tuller, Katherine E. A. Semrau, Linda Vesel.

**Supervision:** Irving Hoffman, Karim Manji, Friday Saidi, Christopher R. Sudfeld, Sunil S. Vernekar, Rodrick Kisenge, Sarah Somji, Tisungane Mvalo, Roopa Bellad, Sangappa Dhaded, Chaya K. A., Bhavana Koppad, Shilpa Nabapure, Saumya Nanda, Bipsa Singh, S. Yogeshkumar.

**Visualization:** Bethany A. Caruso, Uriel Paniagua.

**Writing – original draft:** Bethany A. Caruso.

**Writing – review & editing:** Uriel Paniagua, Irving Hoffman, Karim Manji, Friday Saidi, Christopher R. Sudfeld, Sunil S. Vernekar, Mohamed Bakari, Christopher P. Duggan, George C. Kibogoyo, Rodrick Kisenge, Sarah Somji, Eddah Kafansiyanji, Tisungane Mvalo, Naomie Nyirenda, Melda Phiri, Roopa Bellad, Sangappa Dhaded, Chaya K. A., Bhavana Koppad, Shilpa Nabapure, Saumya Nanda, Bipsa Singh, S. Yogeshkumar, Katelyn Fleming, Krysten North, Danielle E. Tuller, Katherine E. A. Semrau, Linda Vesel, Melissa F. Young.

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
