## [Decision Letter · Decision Letter 0]

19 Jan 2023

PGPH-D-22-01747

Safe infant feeding in healthcare facilities: Assessment of infection prevention and control conditions and behaviors in India, Malawi, and Tanzania

Dear Dr. Caruso,

Thank you for submitting your manuscript to PLOS Global Public Health. After careful consideration, we feel that it has merit but does not fully meet PLOS Global Public Health’s publication criteria as it currently stands. Therefore, we invite you to submit a revised version of the manuscript that addresses the points raised during the review process.

In the methods, please clarify if an existing tool was used or adapted to guide data collection for the facility needs assessment (e.g., WHO IPC Readiness Assessment tool). Please additionally present findings by country and by facility in the results, as this could inform more contextually-targeted recommendations. Finally, please discuss environmental hygiene in the context of resource limitations in LMICs, highlight the need for formal guidance and policy around this topic, and provide specific recommendations for generating further evidence in this area.

We look forward to receiving your revised manuscript.

Kind regards,

Melissa Morgan Medvedev, M.D., Ph.D.

Academic Editor

Journal Requirements:

2. Please provide additional details regarding ethical approval in the body of your manuscript. In the Methods section, please ensure that you have specified the name of the IRB/ethics committee(s) that approved your study.

3. Please include a complete copy of PLOS’ questionnaire on inclusivity in global research in your revised manuscript. Our policy for research in this area aims to improve transparency in the reporting of research performed outside of researchers’ own country or community. The policy applies to researchers who have travelled to a different country to conduct research, research with Indigenous populations or their lands, and research on cultural artefacts. The questionnaire can also be requested at the journal’s discretion for any other submissions, even if these conditions are not met.  Please find more information on the policy and a link to download a blank copy of the questionnaire here: https://journals.plos.org/globalpublichealth/s/best-practices-in-research-reporting. Please upload a completed version of your questionnaire as Supporting Information when you resubmit your manuscript.

4. Please amend your detailed Financial Disclosure statement. This is published with the article. It must therefore be completed in full sentences and contain the exact wording you wish to be published.

Additional Editor Comments (if provided):

Reviewers' comments:

Reviewer's Responses to Questions

**Comments to the Author**

1. Does this manuscript meet PLOS Global Public Health’s publication criteria? Is the manuscript technically sound, and do the data support the conclusions? The manuscript must describe methodologically and ethically rigorous research with conclusions that are appropriately drawn based on the data presented.

Reviewer #1: Partly

Reviewer #2: Partly

Reviewer #3: Yes

2. Has the statistical analysis been performed appropriately and rigorously?

Reviewer #1: N/A

Reviewer #2: No

Reviewer #3: Yes

3. Have the authors made all data underlying the findings in their manuscript fully available (please refer to the Data Availability Statement at the start of the manuscript PDF file)?

Reviewer #1: Yes

Reviewer #2: Yes

Reviewer #3: Yes

4. Is the manuscript presented in an intelligible fashion and written in standard English?

Reviewer #1: Yes

Reviewer #2: Yes

Reviewer #3: Yes

5. Review Comments to the Author

Reviewer #1: Safe infant feeding in healthcare facilities: Assessment of infection prevention and control conditions and behaviors in India, Malawi, and Tanzania

Overall, this is an interesting article addressing a critical information gap. My primary questions / critiques relate to the methods and presentation of the results.

Introduction: The introduction is good, but covers a wide range of topics and feels a bit unwieldy at times. Consider consolidating / restructuring the first two paragraphs to focus in on the issues presented in the manuscript?

Data collection And Analysis:

- what informed the facility needs assessment, beyond the JMP standard questions? There are various tools available - such as the WHO IPC Readiness Assessment - that could guide this data collection. If an existing tool was used or adapted, it would be useful to include that in the methods section. Further, only the classification of WASH services is presented - what was done with the rest of the data?

- Related: WASH data is only reported for water and sanitation. What about hygiene? If data on hygiene facilities were not collected, consider referring replacing “WASH” with “water and sanitation”.

P8, line 160: Glove use and the added prompt. It’s unclear from the text if feed preparers were or were not supposed to use gloves? Gloves aren’t mentioned in the conceptual model and the text is ambiguous about how this would be incorporated? Could this be combined with the hand washing to be a “YELLOW” if not combined with HWWS (and building on traffic light system)?

Conceptual model: Unclear why both treatment and improved water source are included as two separate variable - if water is safe why is additional treatment required. I can make assumptions here, but I think it would be beneficial to make the logic here explicit.

P8: How long were data collectors in each facility? Given the instructions to observe 5 feed preparations and that only 2 facilities reached this number and three facilities recorded no feed preparations - a bit more information is needed to understand the sampling here. Were feeding observations done simultaneously with other data collection? How many data collectors per facility? Where were data collection staff located etc?

Page 9, line 189: Facilities were labeled A - L (not F). Consider breaking at line 184 for a new paragraph and starting this new paragraph with “for all data..” Or something to indicate that this was the approach used for all data collected, not just feeding observations.

Direct observation (page numbering resets after the table, so page numbers below refer to the third numbering):

Why were no feeding preparations observed in facilities J - L?

P2, last paragraph: Authors report a total of 27 observations completed. Page 4 - the denominator is reported as 28. Not sure where the extra observation came in?

P3: suggest moving Table 2 to earlier (maybe just here for page break reasons, but it was a bit confusing to suddenly jump back to previously read data)

P5, first paragraph: this paragraph uses the word reported. Were these practices not observed?

There’s quite a bit of missing data - table 2 combines “missing data” with “not observed” but these are two very different things. Would be useful to differentiate between the two when possible.

Discussion:

In the end - what’s the contribution of the sanitation data? It’s only in the tables and it’s importance is not reflected back in the discussion. Considering all of the facilities had both improved sanitation facilities and improved water sources - what is the data contributing to the manuscript?

Reviewer #2: Congratulations to the team, the information adds value to change of practice and behaviours towards safe infant feeding in our health facilities.

Authors are contributing to an important piece of baseline information alerting more actions needed

The following are minor comments which need to be addressed

ABSTRACT

Well written

METHODOLOGY

Study setting

Line 122 – 124 states that, The LIFE study was conducted in 12 secondary and tertiary level facilities (both public and private) in four urban sites across three countries: (1) Karnataka and (2) Odisha states in India;(3) Lilongwe, Malawi; and (4) Dar es Salaam

The 4 urban sites are confusing, I suggest to just mention 3 countries, I presume that all 12 facilities were in urban settings.

Add rationale of including public and private sites

ANALYSIS

General comments

It will be useful to analyse and show results per country and per facility when needed. Results may have been pulled to certain facilities.

Analyse of public Vs private – is key to explore on any differences or learnings

Observations analysis per facility per country is important – this may lead to country specific recommendations/observations rather than general.

Figure 2 – couldn’t be read, the font is too small and unreadable

Other comments

Among 270 behaviours assessed across the 27 feeding preparation observations, 46

(17.0%) practices were carried out sub optimally, including preparers not handwashing prior to

preparation, and cleaning, drying, and storing of feeding implements in ways that do not

effectively prevent contamination

What were other behaviours observed to be suboptimal?? Only hand washing was discussed

Line 179 -181: ‘We calculated the frequency of the various WASH-related practices for feed preparations overall and by facility. For each observation, we also determined which practices carried out as part of the feeding preparations were optimal and which were suboptimal based on the guidelines’

This was not pictured well in the results (Table or Figure), may be its in Figure 2 – which is difficult to view

Fig. 2. Traffic Light Visual Identifying Optimal and Suboptimal Practices Within and Across Facilities Based on Direct Observations.

Limitation

Hawthorn effect

Line 172 – 173 reads ‘Data collectors carried out observations opportunistically as they noticed a feed being prepared for any baby in the facility’

What does opportunistically mean?? How was hawthorn effect controlled?

What was the interobserver variability?

Distribution of observations per country? Per facility?

Reviewer #3: This is a novel and useful study on infection prevention and control conditions and behaviors and the relationship and infant feeding provision in healthcare facilities across multiple study sites.

I do not have any major concerns with the manuscript, and believe it sound and of value.

Please see attached a few comments which might be useful (or not!) to perhaps expand thinking and tweak the manuscript slightly.

My main point is I would discuss environmental hygiene and limitations of what we expect given resource availability; also to highlight further the lack of formal guidance and policy around this topic and to emphasise the need for this and how we might go about generating further evidence towards formal policy, how to achieve etc.

Thank you.

6. PLOS authors have the option to publish the peer review history of their article (what does this mean?). If published, this will include your full peer review and any attached files.

**Do you want your identity to be public for this peer review?** For information about this choice, including consent withdrawal, please see our Privacy Policy.

Reviewer #1: No

Reviewer #2: No

Reviewer #3: No

---

## [Decision Letter · Decision Letter 1]

31 Mar 2023

Safe infant feeding in healthcare facilities: Assessment of infection prevention and control conditions and behaviors in India, Malawi, and Tanzania

PGPH-D-22-01747R1

Dear Dr Caruso,

We are pleased to inform you that your manuscript 'Safe infant feeding in healthcare facilities: Assessment of infection prevention and control conditions and behaviors in India, Malawi, and Tanzania' has been provisionally accepted for publication in PLOS Global Public Health.

Best regards,

Melissa Morgan Medvedev, M.D., Ph.D.

Academic Editor

Reviewer Comments (if any, and for reference):

Reviewer's Responses to Questions

**Comments to the Author**

1. If the authors have adequately addressed your comments raised in a previous round of review and you feel that this manuscript is now acceptable for publication, you may indicate that here to bypass the “Comments to the Author” section, enter your conflict of interest statement in the “Confidential to Editor” section, and submit your "Accept" recommendation.

Reviewer #2: All comments have been addressed

Reviewer #3: All comments have been addressed

2. Does this manuscript meet PLOS Global Public Health’s publication criteria? Is the manuscript technically sound, and do the data support the conclusions? The manuscript must describe methodologically and ethically rigorous research with conclusions that are appropriately drawn based on the data presented.

Reviewer #2: Yes

Reviewer #3: Yes

3. Has the statistical analysis been performed appropriately and rigorously?

Reviewer #2: Yes

Reviewer #3: Yes

4. Have the authors made all data underlying the findings in their manuscript fully available (please refer to the Data Availability Statement at the start of the manuscript PDF file)?

Reviewer #2: Yes

Reviewer #3: Yes

5. Is the manuscript presented in an intelligible fashion and written in standard English?

Reviewer #2: Yes

Reviewer #3: Yes

6. Review Comments to the Author

Reviewer #2: This is important work creating a foundation to improve breast feeding practices for quality care.

Reviewer #3: Good work implementing changes from all reviewers. I believe the manuscript is improved and of good quality.

7. PLOS authors have the option to publish the peer review history of their article (what does this mean?). If published, this will include your full peer review and any attached files.

**Do you want your identity to be public for this peer review?** For information about this choice, including consent withdrawal, please see our Privacy Policy.

Reviewer #2: **Yes: **Nahya Salim Masoud

Reviewer #3: No
